



# 3D assimilation and radiative impact assessment of aerosol black carbon over the Indian region using aircraft, balloon, ground-based, and multi-satellite observations

Nair K. Kala[1,2], Narayana Sarma Anand[3], Mohanan R. Manoj[2], Srinivasan Prasanth[2], Harshavardhana S. Pathak[2],
Thara Prabhakaran[4], Pramod D. Safai[4], Krishnaswamy K. Moorthy[2] and Sreedharan K. Satheesh[1, 2, 5]

[1]Centre for Atmospheric and Oceanic Sciences, Indian Institute of Science, Bengaluru, Karnataka, India

[2]Divecha Centre for Climate Change, Indian Institute of Science, Bengaluru, Karnataka, India

[3]School of Physics, Indian Institute of Science Education and Research Thiruvananthapuram, Kerala, India

[4]Indian Institute of Tropical Meteorology (IITM), Ministry of Earth Sciences, Pune, Maharashtra, India

[5]DST-Centre of Excellence in Climate Change, Indian Institute of Science, Bengaluru, Karnataka, India

Correspondence: Nair K. Kala (kalanair56@gmail.com)

**Abstract.** A three-dimensional (spatial and vertical) gridded data set of black carbon (BC) aerosols has been developed for the first time over the Indian mainland using data from a dense ground-based network, aircraft- and balloon-based measurements from multiple campaigns, and multi-satellite observations, following statistical assimilation techniques. The assimilated data reveals that the satellite products tend to underestimate (overestimate) the aerosol absorption at lower (higher) altitudes with possible climate implications. The regional maps of atmospheric heating due to BC, derived using this dataset, well-captures the elevated aerosol heating layers over the Indian region and the spatial high over the Indo Gangetic Plains. It is shown that over most of the Indian region, the incorporation of realistic profiles of aerosol absorption/extinction coefficients and SSA into the radiative transfer calculations leads to enhanced high-altitude warming. This will have larger implications for atmospheric stability than what would be predicted using satellite observations alone and could strongly influence the upper tropospheric and lower stratospheric processes, including increased vertical transport of BC to higher altitudes. The 3D assimilated BC data set will be helpful in reducing the uncertainty in aerosol radiative effects in climate model simulations over the Indian region.

**Keywords**: Black Carbon, 3D-Var Assimilation, CALIOP, Radiative Transfer, High-altitude warming

## 1. Introduction

Among the various atmospheric aerosol species, Black Carbon (BC), due to its wide-band absorption of solar radiation (Bond et al., 2013) and large spatial and temporal heterogeneity (Williams et al., 2022), plays an important role in climate forcing (Schmidt and Noack, 2000; Forbes et al., 2006; Bond et al., 2013). BC, mainly produced through low-temperature combustion (anthropogenic activities such as transport, open burning, industrial emissions etc. and natural processes such as forest fires), gets transported to higher altitudes (reaching as high as the stratosphere; Torres et al., 2020) and distant continents and polar regions (Chin et al., 2007; Gogoi et al., 2016; Sicard et al., 2019). It can influence atmospheric stability (Babu et al., 2011), circulation (Lau et al., 2006), and air quality (Aneberg et al., 2011). Though the columnar loading of aerosols, represented using the aerosol optical depth (AOD), is a good parameter to infer the effects of aerosols on the radiation budget, accurate estimation of the absorption potential of aerosols is essential to quantify the vertical structure of aerosol-induced



atmospheric warming and its implications for climate. This happens because the same AOD values can lead to different radiative forcing by aerosols based on the single scattering albedo (SSA), lifetime, and altitude of the aerosols.

40        BC has a longer lifetime in the atmosphere than other aerosol species due to its fine size range and chemically inert nature (Babu and Moorthy, 2002) and gets transported faraway from its sources (Ogren and Charlson, 1983; Chuang et al., 2002; Masiello, 2004). The transport of BC is controlled by meteorological conditions, mainly the wind speed and atmospheric boundary layer (ABL) height (Babu and Moorthy, 2001; Bond et al., 2013). Due to its heating effect, BC can impact the stability of the atmosphere and form stable layers curbing

convection (Wang et al., 2007; Babu et al. 2011). The heating effect of BC depends on its concentration and its vertical distribution. Considering the adverse impact of BC on global climate, efforts are being made to reduce its emission and mitigate global warming (Grieshop et al., 2009; Shindell et al., 2012). Regionally, the radiative impacts of BC are higher over the Indian mainland and the surrounding oceans compared to the global estimates (Babu et al., 2004, Wang et al., 2007). BC is known to alter the precipitation patterns due to its induced heating

at higher altitudes and varying spatial patterns. This effect gains more importance over the South Asia and the Indian region (Menon et al., 2002; Samset et al., 2019; Williams et al., 2022), which heavily depends on the summer monsoon for meeting its water supply. It can thus have far-reaching consequences on the onset of monsoon (and thereby the socio-economic well-being), flood/droughts, weather forecasting and cyclogenesis. Hence an immediate requirement to identify and quantify the absorption potential of aerosols over the Indian

region and delineate their seasonal as well as spatial variations emerge clearly.

        BC over the Indian region is produced from various sources like domestic burning, vegetation wildfires, crop residue burning and fossil fuel combustion (vehicles, industry, etc.). While fossil fuel-based emissions dominate in the urban regions, biomass burning is common in the rural areas (Dey and Di Girolamo, 2010; Rehman et al., 2011; Yadav 2014; Srivastava et al., 2014; Mor et al., 2016). Though wildfires, known to

emit a significant amount of BC, are not as frequent or intense over India (Venkataraman et al., 2006; Kharol and Badarinath, 2006; Bali et al., 2017) as in the other parts of the world, the shifting cultivation practiced in the north-eastern part of India (Badarinath et al., 2004) contribute significantly to BC emissions. The increase in BC loading is shown to be partly leading to the intensification of tropical cyclones in the Arabian Sea (Evan et al., 2011). Furthermore, there is a considerable variation in the vertical distribution of BC and its subsequent radiative forcing

all over the Indian region (Manoj et al., 2020, Ratnam et al., 2021). It has been shown that the radiative forcing increases significantly in the presence of highly reflective clouds and hence the forcing due to BC critically depends on the vertical distribution of BC with respect to low-altitude clouds (Haywood and Shine et al., 1997; Satheesh et al., 2002; Zarzycki and Bond, 2010; Samset and Myhre, 2011). On regional scales, the elevated layers of BC aerosols can even reverse the scattering effect at the top of the atmosphere by reducing the cloud cover

through the atmospheric heating induced at higher altitudes (Ackerman et al., 2000; Keil et al., 2001; Babu et al., 2011; Govardhan et al., 2017).

        However, the assessment of spatial and vertical distribution of BC over the Indian region is challenging due to its diverse geography, multiple emission sources, growing anthropogenic activity and seasonally varying meteorological features. Taking this into account, a network of ground-based observatories (ARFINET, Aerosol



Radiative Forcing over India NETwork; Moorthy et al., 2013) has been established over the Indian region. Besides other aerosol parameters, continuous measurements of near-surface BC mass concentrations are made from about 40 ground-based stations scattered across the mainland (Moorthy et al., 2013; Manoj et al., 2019). In addition to these continuous measurements, several campaign mode measurements have been conducted to fathom the vertical distribution of BC over India (Babu et al., 2010; 2011; 2016; Safai et al., 2012; Vaishya et al., 2018).

These observations, though limited in the spatial and temporal coverage, have shown the persistence of elevated BC layers over different parts of India (Satheesh et al., 2008; Babu et al., 2011; Padmakumari et al., 2013, Vaishya et al., 2018, Manoj et al., 2020). It is known that the in-situ measurements using aircraft and high-altitude balloon, provides a more reliable estimate of the distribution with higher accuracy (Babu et al., 2008; 2011). Conducting periodic campaign mode measurements is not practical, and hence data from such campaigns are sparse, and

spatially and temporally inhomogeneous.

        Although the existing measurements provide a good representation of the BC distribution over the Indian region, there are large discontinuities (both spatially and temporally) and does not provide a homogeneous gridded data set needed for inputting to models, which calls for the assimilation of in-situ measurements with satellite-derived data. This has been carried out globally at regional scales, to get improved representations of the BC

distribution (Benedetti and Fisher 2007; Kahnert et al., 2008; Morcrette et al., 2008; Benedetti et al., 2009; Rouïl et al., 2009; Schutgens et al., 2010; Pagowski et al., 2010; Liu et al., 2011; Pagowski and Grell 2012; Schwartz et al., 2012; Saide et al., 2013; Schwartz et al., 2014). Over the Indian region, Pathak et al. (2019) have constructed gridded Aerosol Absorption Optical depth (AAOD), which accounts for the absorption by BC and dust aerosols. However, these assimilated datasets are column-integrated products and lacks the vital information on their

vertical distribution, which is needed to improve the accuracy of radiative forcing estimates (Clarke et al., 2004). Estimating atmospheric heating rates using column-integrated aerosol properties and height-resolved aerosol properties are shown to differ significantly over oceans as well as landmass (Moorthy et al., 2009, Manoj et al., 2020).

        Unlike other aerosol optical properties, direct quantification of BC through satellite observations is not

possible. Even the BC AAOD data sets derived recently (for e.g., Pathak et al. 2019) provides only columnar information on BC absorption, while the important information on the vertical structure is smoothed off. The present study attempts to fill this important gap by developing an assimilated gridded three-dimensional BC data for the first time over the Indian region, from multi-platform measurement data, following well-accepted statistical assimilation techniques. These profiles are then used to delineate the impact of BC on the thermal profile of the

atmosphere. Sections 2.1, 2.3 and 2.3 respectively describe the ground-based, multi-satellite, and aerial (aircraft and balloon) observations used in this study. The assimilation technique and the radiative transfer calculations are detailed in Sections 2.4 and 2.5. The vertical profiles of BC obtained from the background data and after 3D assimilation are presented in Sections 3.1 and 3.2, respectively. The quality enhancement in the BC distribution over the Indian region, emerging out due to 3D assimilation is explained in Section 3.3. The resulting climate

implications are discussed in Section 3.4. The study is summarized in Section 4 and the important findings are listed out.

        **2.    Data and methodology**



The primary database comprised of those from the dense ground-based network stations, airborne measurements (aircraft and balloon), derived data from multi-satellite observations and reanalysis data. These are used to generate seasonal, 3-D assimilated data set of BC absorption coefficient over the Indian region. For this, the primary data are grouped into the seasons winter (DJF; December to February), pre-monsoon (MAM; March to May), and post-monsoon (ON; October to November) during the assimilation process. The assimilated data sets are thereafter used in the radiative transfer calculations and the improvements are quantified.

### 2.1. Ground-based measurements

A network of observatories (ARFINET), dedicated for aerosol measurements (spectral AOD and near-surface BC), has been set up in a phased manner in the Indian mainland and the islands surrounding it as part of the Aerosol Radiative Forcing over India (ARFI) project under the Geosphere-Biosphere Program (GBP) of the Indian Space Research Organization (ISRO). The geographical locations of the measurement sites are shown in Fig. 1, while the relevant details are listed in Table S1 in the supplementary section. Near-surface BC mass concentration is obtained from Aethalometers (models AE31, AE33, or AE42; Magee Scientific), wherein the ambient air is aspirated following standard protocols and the optical attenuation brought about by the BC aerosols deposited on a quartz filter tape is converted to its mass concentration (Hansen et al., 1984). A detailed description of the operation, data quality check and corrections, and error analysis may be found elsewhere (Babu et al., 2002; Moorthy et al., 2013; Manoj et al., 2019).

### 2.2. Satellite observations

Long-term (2007–2020) data from the spaceborne lidar Cloud-Aerosol Lidar with Orthogonal Polarization (CALIOP) onboard Cloud Aerosol Lidar and Infrared Pathfinder Satellite Observations (CALIPSO) satellite over the Indian region formed the crux of the satellite data employed in this study. Level-2, day and night product (Version 4.20; Young et al., 2018; Liu et al., 2019) at 532 nm wavelength has been used to generate the aerosol extinction coefficient profiles. These profiles were cloud-screened using the Cloud-Aerosol Discrimination (CAD) score by rejecting data points with CAD score outside the range -100 to -80 (Liu et al., 2009). They were further subjected to another screening using the Atmospheric Volume Description (AVD) flag to ensure that the dataset consists of signals from aerosols alone. The aerosol extinction coefficient uncertainty and quality control checks were performed through their respective flags. CALIOP profiles, normalized by the assimilated composite AOD reported by Pathak et al. (2019), formed the background data for 3D assimilation in this study. A rigorous description about CALIOP data screening may be found in Kala et al. (2022) and is not repeated here.

Monthly mean, level-3 aerosol absorption optical depth from Ozone Monitoring Instrument (OMI) aboard the Aura satellite has been used to obtain the absorption AOD due to composite aerosols for 500 nm wavelength (Torres et al., 2005; Pathak et al. 2019). This absorption AOD is the result of AODs mainly due to BC and dust aerosols. To delineate dust absorption, we have used infrared radiance measurements within the wavelength band 10.5–12.5 microns from Very High Resolution Radiometer (VHRR) aboard Kalpana-1 and INSAT-3A satellites (Pathak et al., 2019), which is subtracted from the total absorption AOD to infer AAOD due to BC alone. In addition to this, spectral surface reflectance (Surface Reflectance product Daily L2G Global 250m) values



obtained from Moderate Resolution Imaging Spectroradiometer (MODIS) onboard Aqua and Terra satellites have been used in the radiative transfer calculations in the present study (described in Sec 2.5).

As the primary data from ARFINET do not provide vertical profiles, we have inferred the vertical profiles of BC by first generating BC absorption aerosol optical depth (AAOD) at the ARFINET stations and then normalizing the CALIOP aerosol extinction coefficient profiles using this BC AAOD, with the implied assumption that the vertical variations in composite aerosols and BC are similar over long (for e.g., seasonal) time

scales. BC AAODs at the network stations were evaluated using the Mie scattering model, Optical Properties of Aerosols and Clouds (OPAC; Hess et al., 1998) with the ARFINET BC mass concentration and atmospheric boundary layer (ABL) height obtained from the Modern-Era Retrospective analysis for Research and Applications, Version-2 (MERRA-2; Gelaro et al., 2017) reanalysis as its input. The essential details on the stations and the spatial variations in BC AAOD evaluated at the ARFINET stations are given in the supplementary

section in Table S1 and Fig. S1 respectively. More details on the instrumentation and methodology followed in estimating BC AAOD from near-surface BC measurements may be found in Pathak et al. (2019).

The assimilated AAOD thus obtained (Pathak et al., 2019) provides only columnar values, and no information on its vertical distribution, which is known to be heterogeneous with structures and elevated layers (for e.g., Satheesh et al., 2008; Babu et al., 2011; Kala et al., 2022). Hence, a mere extension of surface BC

measurements to higher altitudes, assuming a constant mixing ratio within the ABL and exponentially decreasing thereafter, will not be realistic over the Indian region. As such, better realistic BC profiles at these locations (where BC AAOD has been evaluated) have been generated by normalizing the CALIOP aerosol extinction coefficient profiles using the BC AAOD shown in Fig. S1 (see Sect. 2.2 for details on CALIOP data analysis). Finally, the balloon and aircraft measurements along with the BC profiles generated from surface measurements and CALIOP

are combined and gridded at 1°×1° spatial resolution and 0.5 km vertical resolution to form the observational input ($k_{obs}$) to the data assimilation, and are shown in Fig. 2.

### 2.3. Aircraft and balloon data

In-situ measurements of the vertical profiles of BC aerosols have been carried out from different locations over the Indian mainland and adjoining oceans during different periods as part of the field various campaigns.

The campaign details are listed in Table 1, and the location from where the aircraft/balloon observations were made are marked in Fig. 1, where the ground-based network stations are also shown. Aircraft measurements were conducted as part of various research campaigns such as the Integrated Campaign for Aerosols, gases and Radiation Budget (ICARB: Moorthy et al., 2006), Winter Integrated Campaign for Aerosols, gases and Radiation Budget (WICARB: Moorthy et al., 2010), multiple phases of Cloud Aerosol Interaction and Precipitation

Enhancement (CAIPEEX: Kulkarni et al., 2012; Safai et al., 2012), and Regional Aerosol Warming Experiment (RAWEX: Babu et al., 2016). The high-altitude balloon measurements conducted from Hyderabad as part of RAWEX (Babu et al., 2011; Moorthy et al., 2016) have also been used. These altogether offered a total of sixty-four profiles of aerosol absorption coefficient ($k_{obs}$) over the study region (obtained during the years 2006 to 2013). However, no measurements could be made during the monsoon season (JJAS: June to September) due to technical

issues.





### 2.4. Three-dimensional assimilation

Several assimilation methods are in use for combining scattered in-situ observations with the gridded background data, such as the successive correction methods (SCM; Kalnay, 2003; Lewis et al., 2006) and weighted interpolation method (WIM) which follows the Cressman method (Cressman et al., 1959). These
assimilation methods are based on a specified radius of interest around the in-situ observation, and the variances are minimised within this radius of interest. Rain-gauge measurements (Mitra et al., 2003; 2009) and AOD data (Chung et al., 2005) have been assimilated using this method. But as the in-situ BC absorption coefficient profiles are less dense in our case (despite the number of stations being about 40), employing the above methods will not be produce a homogenous and continuous assimilated product, useful for inputting to model simulations. As such,
in this study, we have used the 3D-Var (three-dimensional variational) assimilation method (Niu et al., 2008; Zhang et al., 2008), which works on the principle of least-square error minimization. It uses the underlying structure of the background error covariance (unlike other methods of assimilation) making it better suitable for our study. This method has been successfully employed in the past to assimilate AOD (Niu et al., 2008), dust aerosol properties (Zhang et al., 2008) and AAOD (Pathak et al., 2019). We have used it to generate
the final assimilated data as a solution of the minimizer of the objective function J(X) which represents the deviation of assimilated data from its parent data set ($k_{obs}$ and $k_{bg}$).

$$J(X) = 0.5[(X-X_b)^T B^{-1}(X-X_b) + (Z-HX)^T O^{-1}(Z-HX)] \tag{1}$$

where the vectors X (n×1), $X_b$ (n×1), Z (m×1) are respectively the assimilated product, the background data ($k_{bg}$), and observational data ($k_{obs}$), and H (m×n) is the interpolation matrix (maps the background data grid location to
the observation data grid location; Kalnay, 2003; Lewis et al., 2006), B (n×n) is the background error covariance matrix and O (m×m) is the observation error covariance matrix. The terms within the parentheses represent the dimensions of the vectors/matrices as the case is.

Error covariance provides the weights for $k_{bg}$ and $k_{obs}$ during the assimilation and dictates the assimilation pattern in the final product through its covariance structure. Hence, construction of the error covariance matrix
forms the base for the patterns formed in the final assimilated product. The error covariance matrices are symmetric and positive definite matrices. The diagonal terms are the variance for each element of the background and observational data, and the off-diagonal elements provide the covariance. To filter the outliers in $X_b$ ($k_{bg}$), values greater than ($\overline{X}_b + 3\sigma$) are removed, where $\overline{X}_b$ and $\sigma$ are respectively the mean and standard deviation of $X_b$. In the observational error covariance matrix, the diagonal elements alone were considered, as the observational
data are spatially inhomogeneous, largely separated and hence can be considered uncorrelated (Niu et al., 2008; Zhang et al., 2008; Singh et al., 2017; Pathak et al., 2019). Hence the covariance (off-diagonal elements) in the observational error matrix is zero and the variance (diagonal elements) associated uncertainties for constructing the observational error covariance matrix is set as the instrument uncertainty, which in the case of Aethalometers is within 2% to 5% (Hansen and Novakov, 1990; Babu et al., 2004; Dumka et al., 2010). Now that
the observational error covariance is diagonal, the final assimilated data is completely dependent on the background error covariance for their spatial pattern in the final assimilated product and makes it a vital part of the assimilation process. The background error covariance matrix is constructed using long-term data of BC



absorption coefficient profiles, as detailed in Sect. 2.3. It is ideal to evaluate the background error covariance matrix with the long-term data to ensure a realistic error covariance matrix. Monthly mean background data ($X_b$) is joined for all the years, for instance, if there are 'k' time steps, then the monthly mean background data matrix 'D' will have a dimension of n×k. The climatology of the matrix D would have a dimension of n×1 and is referred to as C. Then the anomaly matrix will be the difference between the time series matrix D and the climatology matrix C as shown below.

$$A(i, j) = D(i, j) - C(i,1) \tag{2}$$

The index 'i' refers to the spatial location and index 'j' refers to the time step. The final background error covariance matrix is constructed using:

$$B = (I/(k-1))(A\,A^T) \tag{3}$$

Here, 'I' is an identity matrix of size n×n. The background error covariance matrix is evaluated separately for DJF, MAM, and ON seasons to evade the bias introduced due to the seasonality in loading of different aerosol species, along with the changes in the meteorological parameters. However, the background covariance matrix would still be deficit and singular. Hence, to make it a full rank matrix, a small value of the order of $10^{-14}$ (rank deficient) is added to the diagonal elements of the background covariance matrix, which is known as the Tikhonov regularisation method (Lewis et al., 2006). This will not change the structure or variance of the final assimilated product. The final equation for estimating the assimilated BC absorption coefficient, which is the minimizer for the objective function, was obtained by solving Eq. 1:

$$[B^{-1} + H^T O^{-1} H]\,X = [B^{-1} X_b + H^T O^{-1} Z] \tag{4}$$

The assimilated product evaluated using 3D-Var is known to have lesser estimates of variance compared to the background or observational data (Kalnay, 2003; Lewis et al., 2006), also leading to smaller uncertainties in the assimilated product. Since the 3D-var is not a constrained model, the final assimilated product may contain negative values of assimilated absorption coefficient ($k_{asm}$), in which case, those specific grids alone would be replaced by the corresponding $k_{bg}$ values. More details on the 3D-Var method for assimilation can be found in Kalnay (2003), Lewis et al. (2006) and Pathak et al. (2019). The essence of the analyses described above are explained below and summarized in the flowchart in Fig. 3.

An assimilated AAOD product (Pathak et al., 2019) was first generated using OMI composite AAOD, BC AAOD at the ARFINET stations, and dust AAOD [calculated within the spectral band 10.5 – 12.5 μm and estimated using the Infrared Difference Dust Index (IDDI; Legrand et al., 2001)]. Dust AAOD was subtracted from this assimilated AAOD to deduce BC AAOD, which served as the background data. The merits of this assimilated AAOD data are that it contains the positive features from in-situ observations (realistic measurements) and satellite observations (wider spatial coverage). Spatial variations in this assimilated BC AAOD over the Indian region for DJF, MAM, ON seasons are shown in Fig. 4.



It can be observed from Fig. 4 that the BC AAOD values are the highest in the IGP region, moderate in central India, and low in the peninsular, northwest, and eastern regions. This is in good agreement with the spatial pattern of BC over the Indian region reported by Beegum et al. (2009). Previous studies have attributed the high aerosol loading over IGP to both natural and anthropogenic emissions (for e.g., Lawrence and Lelieveld, 2010; Kumar et al., 2020). These anthropogenic BC emissions arise mainly from coal-based thermal power plants (Prasad et al., 2006), vehicular emissions, small-scale industrial emissions, burning of litter, and biofuel burning for domestic cooking (Reddy and Venkataraman, 2002; Girolamo et al., 2004). During MAM, the IGP region is influenced by BC produced from agricultural residue burning and forest fires and the continental inflow of polluted aerosols to this region (Weissmann et al., 2005; Singh et al., 2019).

### 2.5. Radiative transfer calculations

The radiative transfer calculations have been carried out using Santa Barbara DISORT Atmospheric Radiative Transfer (SBDART; Ricchiazzi et al., 1998) model for a plane-parallel and vertically inhomogeneous atmosphere using discrete ordinate method (Stamnes et al., 1988). SBDART calculations were carried out for eight radiation streams and considering an atmosphere having a vertical resolution of 0.5 km from the surface to 10 km altitude and a lower resolution thereafter. Vertical structure of atmospheric thermodynamics pertaining to tropical atmospheric conditions were inputted to SBDART. The spectral surface reflectance values within the visible and near-infra red wavelengths for all the seasons, corresponding to all the grid points were obtained from MODIS (see Sect. 2.2). Accuracy of critical inputs such as the aerosol scattering phase function and SSA, is important in improving the accuracy of the estimated aerosol radiative effects. Since we did not have a species-segregated composition of aerosols over the entire study region, aerosol scattering phase function values were obtained from the Mie scattering model OPAC (Hess et al., 1998). SBDART calculations were made at every 5° solar zenith angle intervals and clear-sky conditions for two cases: "without aerosols" and "with aerosols" over the shortwave radiation spectrum. The difference in the layer-wise net radiative forcing between these two cases provide the aerosol radiative forcing ($\Delta F$), which has been used to estimate the aerosol-induced atmospheric heating rate ($\partial T/\partial t$; Liou, 2002).

$$\frac{\partial T}{\partial t} = \frac{g}{C_p} \frac{\Delta F}{\Delta P} \tag{5}$$

where 'g' is the acceleration due to gravity, $C_p$ is the specific heat capacity of air under constant pressure, and P is the atmospheric pressure. Further details on the estimation of $\Delta F$ and $\partial T/\partial t$ using SBDART may be found in Kala et al. (2022).

### 3. Results and discussion

### 3.1. Vertical profiles of BC from background data

As absorption coefficients and its vertical structure are not directly available from satellite observations, we have used a hybrid method employing multi-satellite and ground-based measurements to generate a background data set of BC absorption coefficient profiles. While most of the earlier studies reported high near-surface BC mass concentration during DJF as compared to MAM, we have seen (Fig. 4) that the BC AAOD values



in MAM are comparable to or higher than that observed in DJF. Similar observations were made by Nair et al. (2016) who reported a difference in the seasonal variation of surface BC mass concentrations and the vertically varying aerosol absorption coefficients during spring (MAM). The shallow ABL during DJF (see Fig. S2) confines the aerosols closer to the surface, leading to large surface concentrations, but the consequent partial shielding of vertical mixing leads to a lower load in the free troposphere. CALIOP aerosol extinction coefficient profiles were normalized with assimilated BC AAOD to create background profiles of 3D distribution of BC absorption ($k_{bg}$) over the Indian region. These profiles have been used as the background data set for 3D assimilation of BC absorption coefficient over the Indian region, along with the observational profiles described in Sect. 2.3. The resulting 3D distributions of BC absorption coefficient, shown in Fig. 5 for DJF, MAM, and ON seasons, clearly show that at higher altitudes, BC is present in larger concentration during MAM (middle panel), as compared to DJF for almost the entire study region.

The vertical profiles of $k_{obs}$, generated by combining aircraft- and ground-based measurements for DJF, MAM, and ON seasons and shown in Fig. 2, lend support to the above features. In general, the values of $k_{obs}$ are higher close to the surface and reduces with increasing altitude. Despite this general trend, high values of $k_{obs}$ are seen at higher altitudes (>5 km) over various locations; especially over central, northern, and eastern parts of the mainland, consistent with the multiple earlier observations of elevated BC layers (Moorthy et al., 2004; Babu et al., 2010; Safai et al., 2012; Vaishya et al., 2018; Manoj et al., 2020). During MAM, the prevalence of these elevated, high values of $k_{obs}$ is more frequent while the whole pattern gradually descends during ON. The vertical extent of BC is higher over the IGP and central India, as compared to the coastal regions and peninsular India. This is evident from the high values of $k_{obs}$, at higher altitudes in IGP and central India. The interplay of surface temperatures and the ABL heights create ideal conditions for the vertical lifting of aerosols in the IGP and central India.

Aircraft measurements during ICARB (Babu et al., 2010) have revealed the presence of highly absorbing aerosol layers above Chennai, a coastal location in peninsular India, at an altitude of ~1.5 km. Similar observations of elevated BC layers were also reported by Babu et al. (2011) and Rahul et al. (2014) over Hyderabad in central India and Guwahati in northeast India, respectively. Previous studies (Gautam et al., 2011; Manoj et al., 2020) showed low SSA over central IGP, the magnitude of which increases towards eastern and western IGP. The seasonal variations in the 3D distribution of background data set ($k_{bg}$) corresponding to the seasons DJF, MAM and ON are shown in Fig. 5. The vertical extents and magnitudes of $k_{bg}$ vary spatially, with the highest over the IGP region followed by central India and least over coastal peninsular India, irrespective of the season, and closely resemble the variations in $k_{obs}$. While the features of $k_{obs}$ are in general similar to those of $k_{bg}$, the vertical extents and magnitudes are lower in $k_{bg}$ as compared to $k_{obs}$, which will have a large impact on the radiative transfer calculations.

### 3.2. Vertical profiles of BC from 3D-Var assimilation

Following the procedures detailed in Sect 2.4, $k_{obs}$ and $k_{bg}$ were used as inputs to generate an assimilated 3D distribution of BC absorption coefficient ($k_{asm}$) over the study domain. The resulting spatially homogenous (1°×1° gridded) maps for the three seasons are shown in Fig. 6. While the broad seasonal features of $k_{obs}$, $k_{bg}$, and $k_{asm}$ are similar (such as the highest vertical extent occurring during MAM and the highest BC absorption coefficient



occurring over the IGP), the assimilation of aircraft and balloon data has resulted in features in the assimilated
product differing significantly from its parents. These include:

(i)    the spring-time enhancement in aerosol absorption coefficient over the high-altitude locations in the
       Himalayan foothills
(ii)   higher absorption coefficient over central IGP compared to its eastern and western parts, prior to the onset
       of the Indian monsoon
(iii)  higher concentration of BC (leading to higher values of the absorption coefficient) and its vertical extent
       during the pre-monsoon season

While these are in general conformity to earlier reported features of the vertical variation of BC (for
example, Nair et al 2016; Babu et al., 2016; Vaishya et al., 2018) over different parts of India, are not conspicuous
in the background data.

A close examination of Fig. 6 reveals that the BC absorption is significant mostly below 4 km altitude
during DJF and ON over most of the, if not the entire, Indian mainland, with the highest values between the ground
to about 1 km. This is attributed mainly to the reduced vertical mixing of BC, being suppressed by the shallow
ABL conditions, characteristic to these seasons, driven by the low to very low surface temperature and calm wind
conditions prevailing over the landmass, especially over the northern latitudes (central and northern Indian
mainland where the temperatures drop to near zero and sub-zero values during winter). However, during the pre-
monsoon/ summer season (MAM), the increased heating of land surface by the increasing solar insolation (as the
sun moves to the northern hemisphere) and the resulting strong thermal convection produces enhanced vertical
mixing and consequently aerosols are lofted to altitudes as high as $5 - 6$ km, with the maximum extent over the
inland regions of Central India (Kala et al., 2022). It may be noted that during MAM, the land surface temperature
goes well above 40 °C over most of the mainland. Nevertheless, the absorption coefficient values are the highest
in the IGP region and is attributed to the larger concentration of absorbing BC and dust aerosols here. Signatures
of high-altitude absorbing aerosol layers can be observed in the western region, which is arid/semi-arid and closer
to the Thar desert, persistently under the influence of transported dust (Moorthy et al., 1997; Chinnam et al., 2006;
Banerjee et al. 2019) during all the seasons, particularly during MAM.

**3.3.    Quality enhanced BC data over the Indian region**

The major outcome of the assimilation is that the assimilated products ($k_{asm}$) show higher values in the free
troposphere than the satellite products (background data $k_{bg}$), which tend to limit the absorption to lower altitudes
(see Figs. 5 and 6). This is demonstrated in Fig. 7, where the spatial variations in $k_{asm}$ are shown at two altitudes,
one sliced closer to the surface (1 km) and the other at a higher altitude (3.5 km). For ease of explanation, the two
altitudes (1 and 3.5 km) are hereafter referred to as the 'low' and 'high' altitudes respectively. The spatial
variations in $k_{asm}$ at the low altitude during DJF, MAM, and ON are shown in Fig. 7 a,b,c and the same at the high
altitude in Fig. 7 g,h,i.

To enunciate this, the difference in BC absorption coefficients between the assimilated and background
data sets ($\delta k = k_{asm} - k_{bg}$) are shown in Fig. 7 d,e,f for the low altitude and in Fig. 7 j,k,l for the high altitude. Most



of the regions over the mainland show positive δk values at the high altitude. This underlines the improvement in the spatial and vertical distribution of aerosol absorption brought-in by the assimilation of ground-based measurement data with satellite retrieved data. The high difference between $k_{asm}$ and $k_{bg}$ observed at grid locations even far away from the $k_{obs}$ grid locations, especially at higher altitudes, reveals the advantage of using 3D-Var assimilation, which is not merely confined to a radius of influence around the $k_{obs}$ grid points. The seasonally

highest difference between $k_{asm}$ and $k_{bg}$ is observed during ON, as compared to DJF and MAM. The band of negative δk at the high altitude in IGP during MAM points to the vertical extent and concentration of BC being much higher than that captured by the background data in pre-monsoon summer. On the other hand, at the low altitude (1 km), δk is observed to be negative over the IGP and close to zero or positive in the central and peninsular regions, implying the overestimation of aerosol absorption at the low altitudes, when only satellite-retrieved background data ($k_{bg}$) is used, especially during winter and post-monsoon seasons, for any climate impact

assessment.

To bring out these features clearly over the entire domain, the 3D distribution of δk for different seasons are shown in the supplementary section (Fig. S3). The overestimation of BC at lower altitudes and underestimation at higher altitudes (above the ABL) during all the seasons are clearly visible from the figure. δk is positive at the

high altitudes during all the seasons over the entire Indian region. The reason for the negative δk band observed in IGP region in Fig. 7k emerges out clearly in Fig. S3b. δk shifts to positive values at altitudes above 4 km during MAM which are mainly due to the increased convective lofting of near-surface emissions over the region during summer (Kala et al., 2022). In addition to this, the envelope separating the positive and negative values of δk have seasonal and spatial variations and is observed to be the highest in the inland regions and lower close to both sides.

This is in line with the observation of the seasonally invariant maximum vertical extent of the aerosols over the inland Indian regions (Kala et al., 2022).

To further quantify the quality-enhancement in BC distribution brought about by 3D assimilation, least-square linear regression fits aree constructed for all the seasons between $k_{obs}$ and $k_{bg}$ and between $k_{obs}$ and $k_{asm}$, and are shown in Fig. 8. It clearly emerges out from the figure that $k_{asm}$ offers an improved and near-realistic

distribution of BC aerosols over the Indian region, as can be seen from its closeness to the observational data. There is an improvement in the correlation coefficients from $k_{obs}$ v/s $k_{bg}$ to $k_{obs}$ v/s $k_{asm}$ scatter plots by 31%, 16%, and 35% respectively during DJF, MAM, and ON seasons. The linear regression coefficients have almost doubled during DJF and ON and increased by about 50% during MAM. It can be observed that the seasons with lesser ABL height (DJF and ON) are marked with larger quality enhancement.

## 3.4.    Climate implications

The radiative effects of BC aerosols depend not only on its concentration, but on its SSA, altitude distribution, and the reflectance of the underlying surface (Satheesh et al., 2008; Babu et al., 2011) as well. Same concentration of BC aerosols at higher altitudes will produce higher atmosphere warming as compared to when it is at a lower altitude, owing to the decreasing atmospheric density with increasing altitude. As such, the

underestimation (overestimation) of BC absorption in the background data at high (low) altitudes will have large climate implications. The climate impacts of BC depend largely on its SSA, whose realistic measurements, though crucial, are sparse over the Indian region. While the SSA measurements provided by satellite sensors such as OMI



capture the spatial variations in aerosol absorption to a large extent, their magnitudes have been associated with large uncertainties over the Indian region (Eswaran et al., 2019), which is partly attributed to the assumption of aerosol layer heights (Satheesh et al., 2009; 2010).

A few ground-based measurements, though offer more accurate columnar SSA estimates, are limited and rather provide point values alone. As such, we have constructed 3D maps of aerosol-induced atmospheric heating rate ($\partial T/\partial t$) over the domain using two different sets of data: (i) the background aerosol absorption data ($k_{bg}$) along with vertically invariant (columnar) SSA derived from OMI, and (ii) the assimilated aerosol data ($k_{asm}$) along with vertically varying SSA, constructed by using the CALIOP aerosol extinction coefficient profiles and the assimilated absorption ($k_{asm}$) discussed above (and shown in Fig. 9). The $\partial T/\partial t$ profiles for different seasons, obtained using the first data set, are shown in Fig. 10, and that using the second data set is shown in Fig. 11.

It can be observed from both Fig. 10 and Fig. 11 that in addition to increasingly capturing the high-altitude heating induced by aerosols, the vertical extent and the height of maximum $\partial T/\partial t$ occurs about 0.5–1 km higher when altitude-resolved SSA profiles, rather than the columnar values, are used in the radiative transfer calculations. Seasonally, the highest $\partial T/\partial t$ occurs during MAM, followed by ON and DJF. The high values of $\partial T/\partial t$ at higher altitudes are associated with the high-altitude $k_{asm}$ values, which, though are lower than the near-surface values at the same location, produce comparable (to near-surface values) $\partial T/\partial t$ values at higher altitudes. Such higher heating rates, despite with lower values of $k_{asm}$ at higher altitudes, arise mainly due to the rarer atmosphere there. The difference in $\partial T/\partial t$ between the two cases are shown in Fig. 12. It emerges out clearly from the figure that the incorporation of altitude-resolved absorption/extinction coefficients and SSA data into the radiative transfer computations has resulted in higher warming above the ABL (especially during DJF and ON seasons) over most of the Indian mainland. This will have significant implications for the atmospheric stability and other boundary layer processes than what would be predicted using satellite data alone or even when columnar SSA is added to the satellite observations.

The transport of boundary layer BC aerosols over the IGP, Himalayas, and BoB into the free atmosphere and upper troposphere and lower stratosphere (UTLS) as part of the monsoon circulation, and its consequent impacts have been extensively studied in the past (for e.g., Pusechel et al., 1992; Fadnavis et al., 2017; 2022; Singh et al., 2020; Maloney et al., 2022; Lau and Kim 2022). The high-altitude excess BC (compared to satellite observations) found in our results are hence likely to be reflected as an excess transport of BC to UTLS and thereby influence the monsoon and Indian climate. It could also lead to increased cloud activities in the higher atmosphere as aged BC can act as cloud condensation nuclei. The excess BC (and warming) in the higher altitudes in the North, when lifted to UTLS, especially during MAM, are likely to warm the Tibetan Plateau and thereby increase the Indian summer monsoon rainfall. The excess BC observed in our assimilated data set (compared to satellite data) may possibly amplify the mid-tropospheric warm anomalies and the tropopause cold anomalies (Fadnavis et al., 2017).

Lau and Kim (2022), using MERRA reanalysis, reported a strong, linearly increasing trend in BC concentration over the entire Asian Summer Monsoon (ASM) region over the past four decades and a conspicuous positive trend over the Indian region at 850 hPa height level. They reported an increased transport of carbonaceous aerosols (including BC) from the surface to the UTLS during the ASM. This in turn will act as a feedback to



increase the radiative heating by UTLS BC aerosols and thereby strengthen the elevated heat pump activities (Lau et al., 2006), to further modify the ASM circulation and the height of the tropical tropopause. Self-lofting of upper atmospheric absorbing aerosols through the 'solar escalator process' has been reported by de Laat et al. (2012) and its possible implications for stratospheric ozone are discussed by Satheesh et al. (2013). Our observations
from the 3D assimilated data set, showing the warming anomalies at different heights should be seen in conjunction with these, thereby making it all the more important. Self-lofting alone can take the smoke plumes in the middle/upper troposphere to the UTLS at rates as high as 1 km per day, even in the absence of pyrocumulus convection (Ohneiser et al., 2022). Even though Ohneiser et al. (2022) argued that high values of tropospheric AOD may be involved in such conditions, they also highlighted that wildfire emissions (often associated with
self-lofting) are not always necessary for self-lofting to occur.

BC aerosols in the stratosphere will modify the radiation budgets locally by modifying the stratospheric temperatures and winds (Doglioni et al., 2022). The radiative heating and the consequent self-lofting of BC plumes can also contribute to the formation, expansion and compression of stratospheric circulations. The concentration and radiative properties (particularly the SSA) of the aerosols determine whether they can be sustained in the
model simulations. Doglioni et al. (2022) underestimated the radiative impacts of BC and hence got a shorter lifetime for the plumes. They pointed out that if realistic BC data (vertical profiles and radiative properties) are provided as input, the models would be able to capture it realistically. Diabatic heating by aerosols can contribute to enhanced pressure (and thus a pressure gradient force) in the centre of the smoke plumes, which helps to maintain the circulation.

During the daytime, absorption of solar radiation by BC aerosols leads to the expansion and vertical motion of the plume. During the nighttime, radiative cooling of the plume leads to its downward motion, which though is lesser compared to the upward motion during the daytime, leads to a net ascent of the plume over the course of a few days time. Khaykin et al. (2020) showed that accurately capturing the aerosol-radiation interactions and its dynamics are crucial in simulating the lofting of such aerosol plumes in model simulations. Maloney et al. (2022)
showed that the stratospheric temperatures are sensitive to even a slight increase in BC aerosols. A resulting shift in the stratospheric dynamics can further modify the concentration of Ozone in the stratosphere. The findings from the present study on the increased free-atmospheric BC concentration (compared to satellite observations) and the possibility of them getting lofted to UTLS will hence have implications to future climate predictions. Dedicated modelling studies using this 3D-assimilated BC data set are reqired to closely examine the climate
implications.

### 4.    Conclusions

A gridded, homogeneous (spatially and vertically), seasonal data of aerosol absorption over the Indian mainland has been constructed, for the first time, by assimilating (employing 3D-Var assimilation technique) dense and continuous near-surface measurements, using multi-campaign in-situ measurements from aircraft and
high-altitude balloons, space-borne lidar measurements and assimilated BC AAOD data. The assimilated dataset offers BC absorption coefficient profiles at a horizontal resolution of $1°×1°$ and a vertical resolution of 0.5 km for three seasons: winter (DJF), pre-monsoon summer (MAM) and post-monsoon (ON). The resulting 3D variation of the radiative effects of BC aerosols is evaluated in terms of aerosol-induced atmospheric heating rates. It is



observed from the assimilated BC data set that the satellite measurements tend to underestimate (overestimate) the aerosol absorption at lower (higher) altitudes. Our findings demonstrate that incorporating altitude-resolved aerosol absorption/extinction coefficients and SSA data into the radiative transfer computations results in higher warming in the free-troposphere (especially during DJF and ON seasons) over most of the Indian region. This enhanced warming at higher altitudes will have large implications for the atmospheric stability, other boundary layer processes and upper troposphere and lower stratosphere (UTLS) transport, than what would be predicted using satellite data alone or even when columnar SSA is added to the satellite observations. The above findings clearly reveal that the assimilated data set ($k_{asm}$) will help reduce the uncertainty in aerosol radiative effects in climate model simulations over the Indian region.

**Data availability**

Processed data are available upon request.

**Author contribution**

NKK, SKS, and KKM together conceived the work. NKK carried out the scientific data analysis and along with NSA, KKM, and SKS were involved in the scientific interpretation of the results, leading to the formulation of the manuscript. NKK prepared the initial draft with inputs from NSA. All authors reviewed the manuscript.

**Competing interests**

The contact author has declared that neither they nor their co-authors have any competing interests.

**Acknowledgements**

CALIOP data used in this study were obtained from the NASA Langley Research Center Atmospheric Science Data Center. MODIS level 2 surface reflectance was acquired from https://ladsweb.nascom.nasa.gov/. Ground station BC data from the ARFINET has been used in this study. We acknowledge the mission scientists of Kalpana-1, INSAT-3A, MODIS and OMI and the associated ISRO and NASA personnel for the data used in this work. We acknowledge NASA Global Modeling and Assimilation Office (GMAO) for the MERRA-2 reanalysis data. This work is partially supported by MoES (grant no. MM/NERC-MoES-1/2014/002) under the South West Asian Aerosol Monsoon Interactions (SWAAMI) project. One of the authors (S. K. Satheesh) was supported by the Tata Education and Development Trust and the J. C. Bose Fellowship awarded by the Science and Engineering Research Board-Department of Science and Technology (SERB-DST). Anand acknowledges the DST for the INSPIRE Faculty Fellowship (faculty registration number: IFA20-EAS-83). We thank Divecha Centre for Climate Change for supporting this work.

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



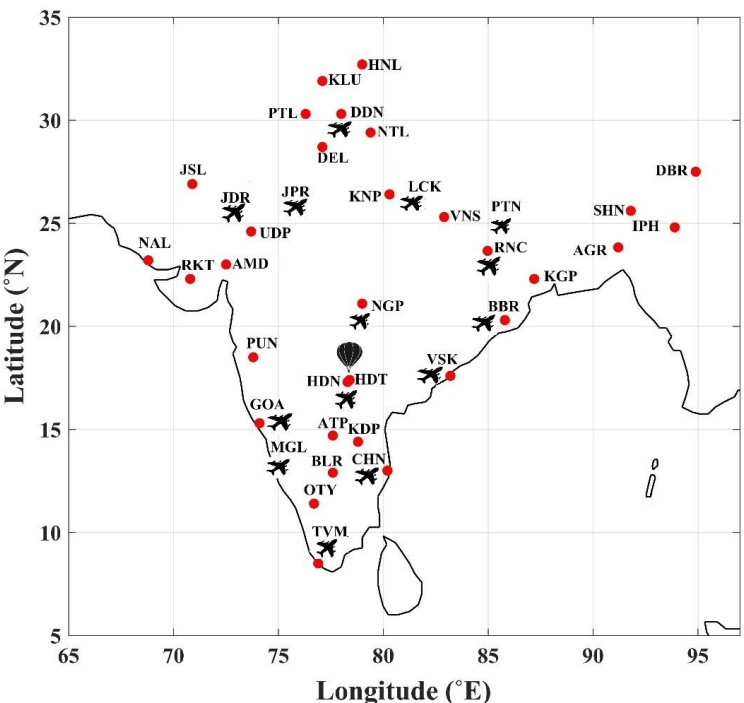

**Figure 1.** Map showing the locations of surface (red markers) and aerial (black markers) measurements. The aircraft symbols mark the aircraft measurements, with the tip of the symbol pointing towards the base station. The balloon symbol marks the high-altitude balloon measurement site. The station codes for the ARFI measurement sites are given in Table 1 and S1.

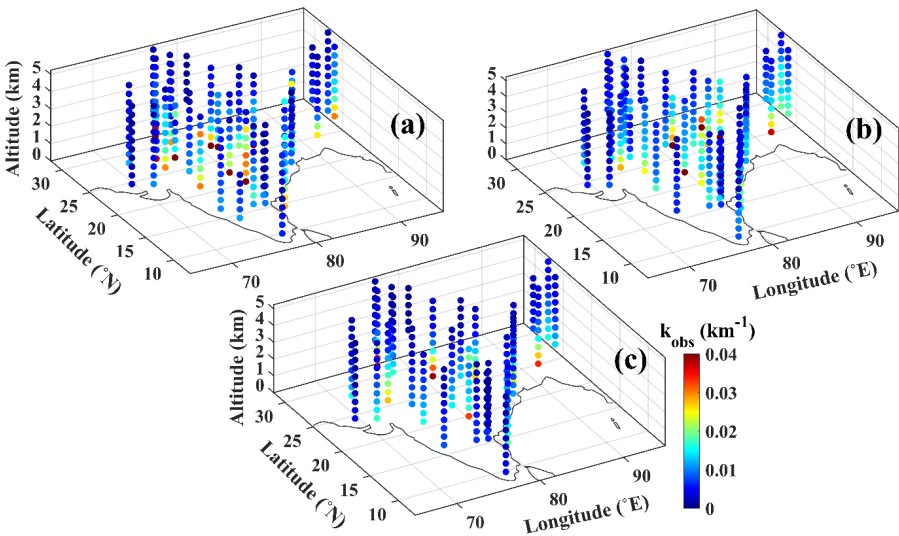

**Figure 2.** Vertical variation of $k_{obs}$ during (a) DJF, (b) MAM, and (c) ON seasons.



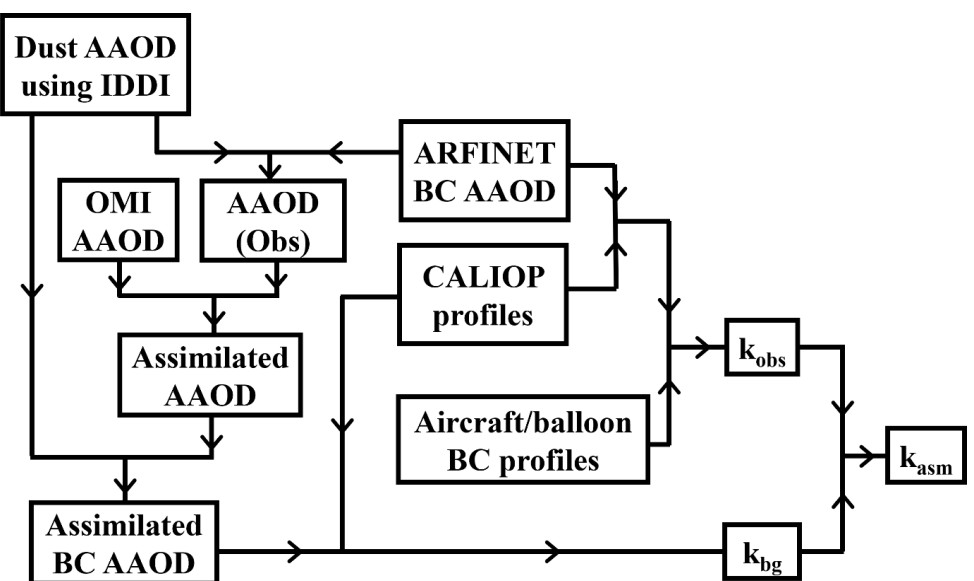


**Figure 3.** Flowchart describing the various data sets and steps involved in the data assimilation. AAOD (Obs) represents the AAOD derived from surface measurements. $k_{obs}$, $k_{bg}$, and $k_{asm}$ respectively represent the observational, background and assimilated aerosol absorption coefficients.

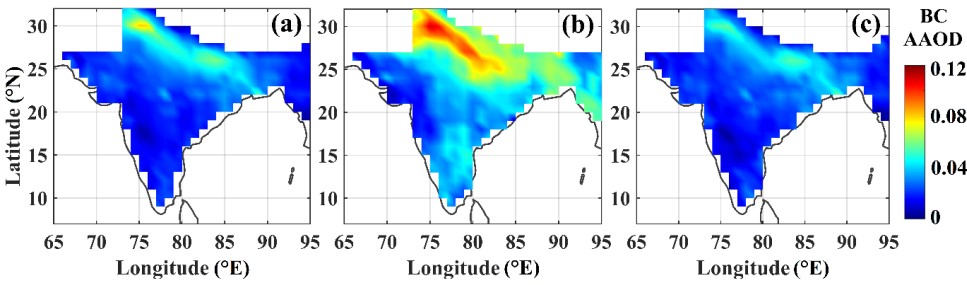

**Figure 4.** Assimilated BC AAOD over the Indian region during (a) DJF, (b) MAM, and (c) ON seasons.



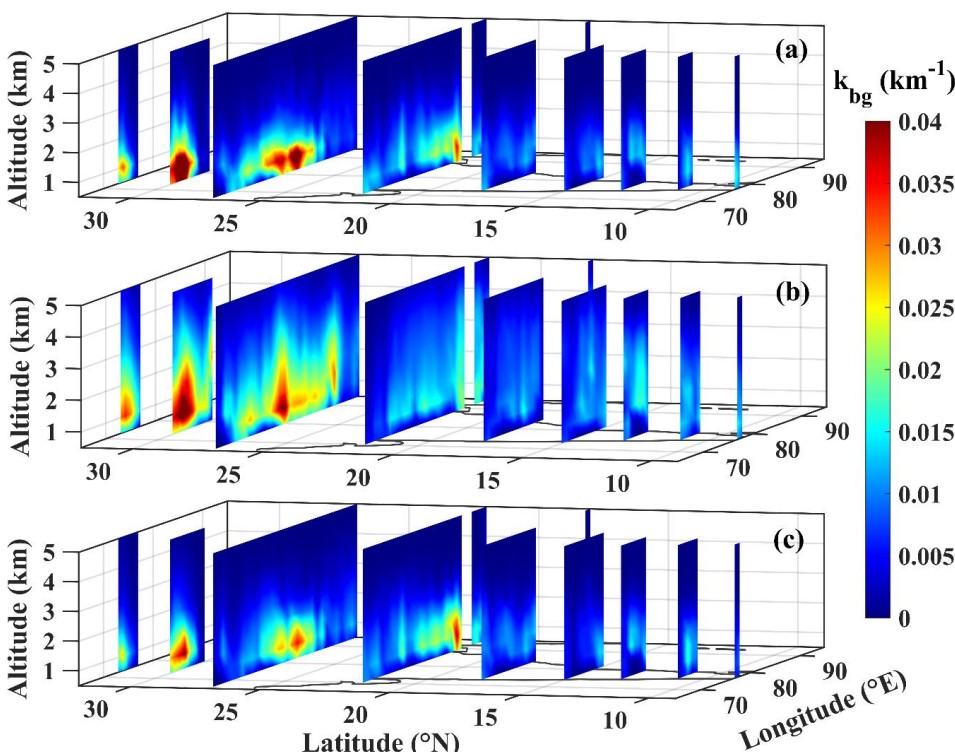

**Figure 5.** The spatial variations in the background BC absorption are represented by the variations in $k_{bg}$ profiles during (a) DJF, (b) MAM, and (c) ON seasons.



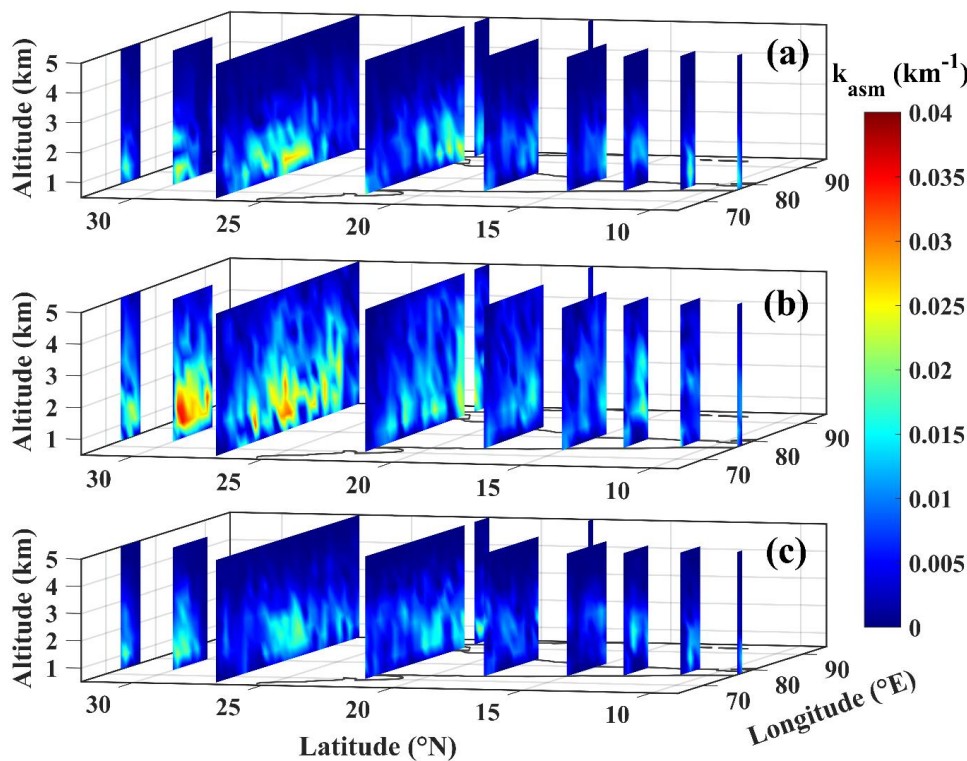

**Figure 6.** Spatial variation of $k_{asm}$ profiles generated by assimilating $k_{bg}$ profiles (Fig. 5) and $k_{obs}$ profiles (Fig. 2) for (a) DJF, (b) MAM, and (c) ON seasons.



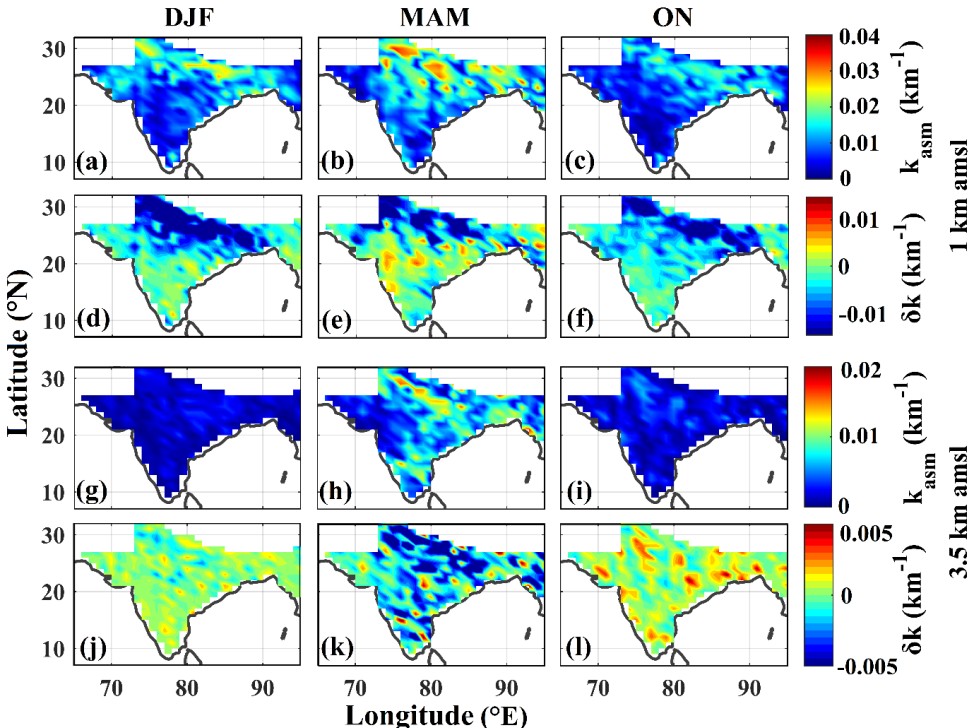

**Figure 7.** Spatial variation of $k_{asm}$ and **δk** at two altitudes (1 km and 3.5 km) for DJF (left panels), MAM (middle panels), and ON (right panels) seasons.




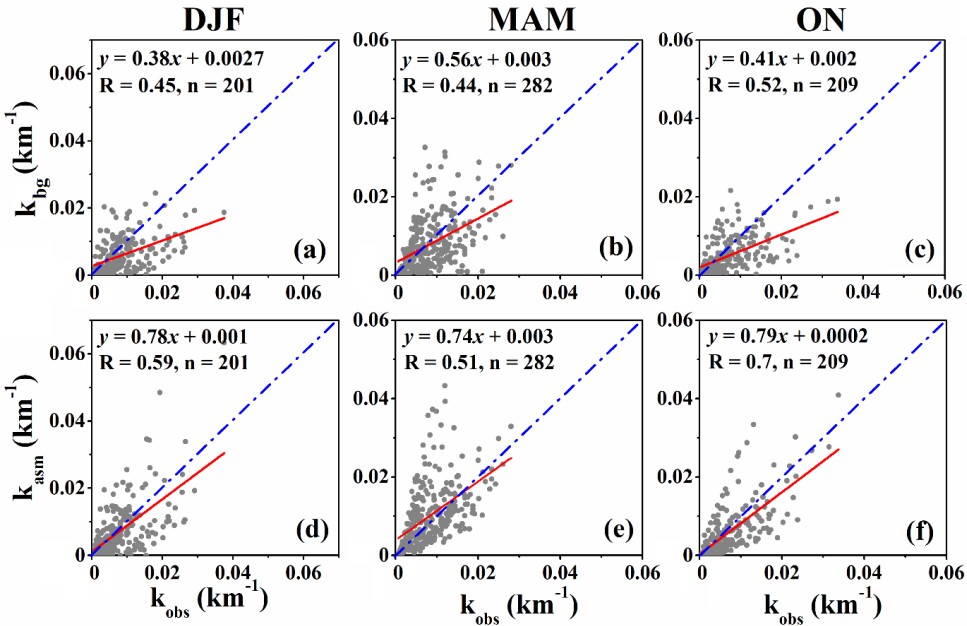

**Figure 8.** Scatter plots between profiles of $k_{obs}$ and $k_{bg}$ (top panels) and between $k_{obs}$ and $k_{asm}$ (bottom panels) for DJF (left panels), MAM (middle panels), and ON (right panels) seasons. The red line denotes the linear fit, the dashed blue line denotes the 1:1 line, and the scatter points are shown in gray. The equation of fit, correlation coefficient (R), and the number of scatter points (n) are shown in each sub plots.




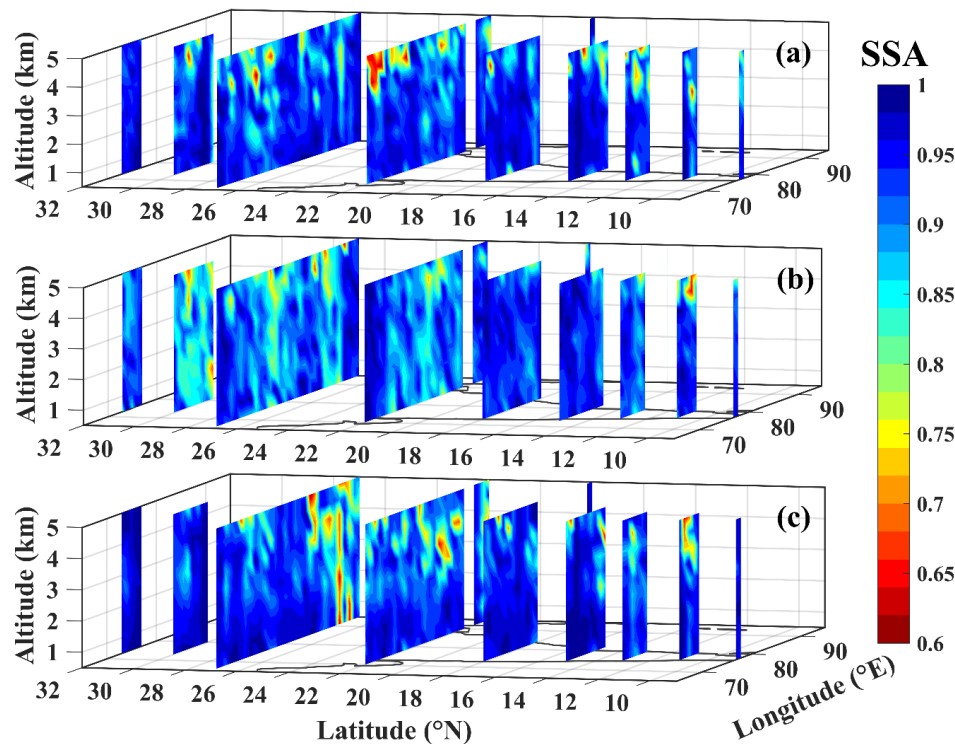

**Figure 9.** SSA profiles for (a) DJF, (b) MAM, and (c) ON seasons generated using CALIOP aerosol extinction coefficient profiles (weighted with assimilated AOD) and assimilated BC absorption coefficient ($k_{asm}$) profiles.



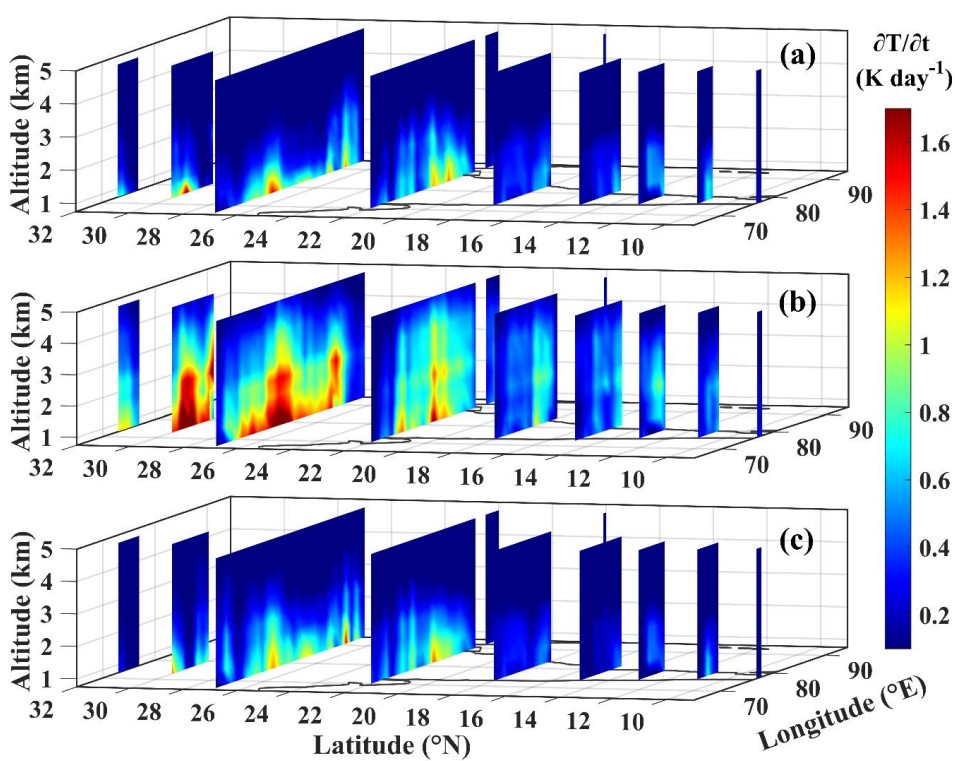

**Figure 10.** Diurnally averaged aerosol-induced atmospheric heating rate ($\partial T/\partial t$) profiles for (a) DJF, (b) MAM, and (c) ON seasons, estimated using background data (CALIOP extinction + columnar SSA).



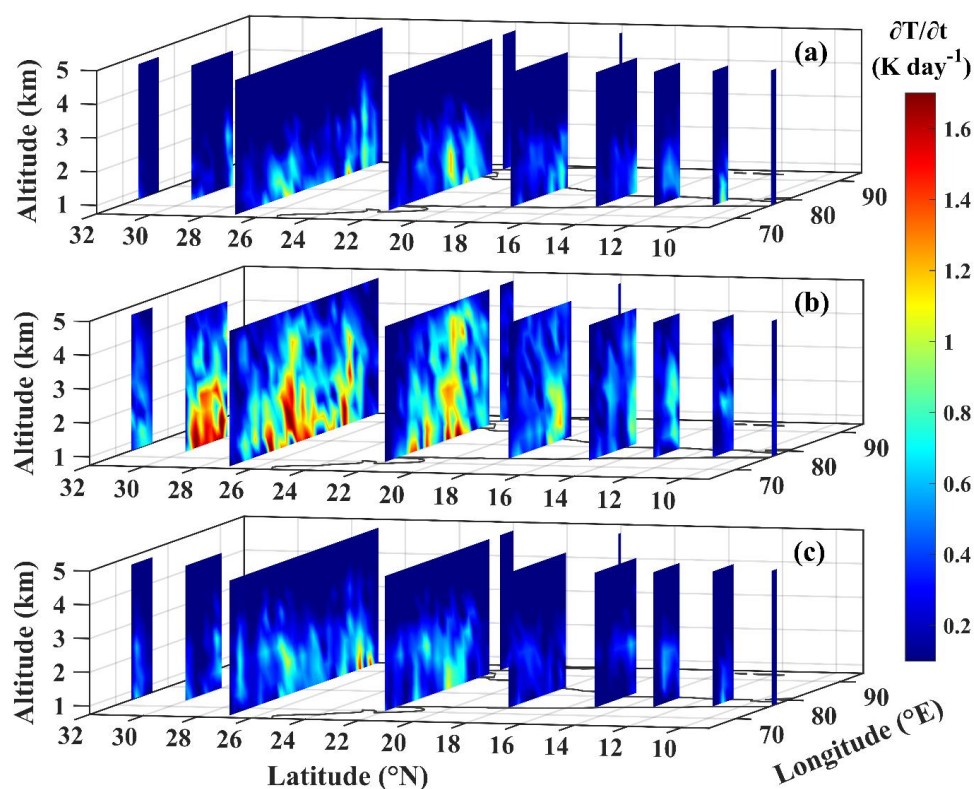

**Figure 11.** Diurnally averaged aerosol-induced atmospheric heating rate ($\partial T/\partial t$) profiles for (a) DJF, (b) MAM, and (c) ON seasons, estimated using assimilated data (CALIOP extinction + realistic SSA profiles).



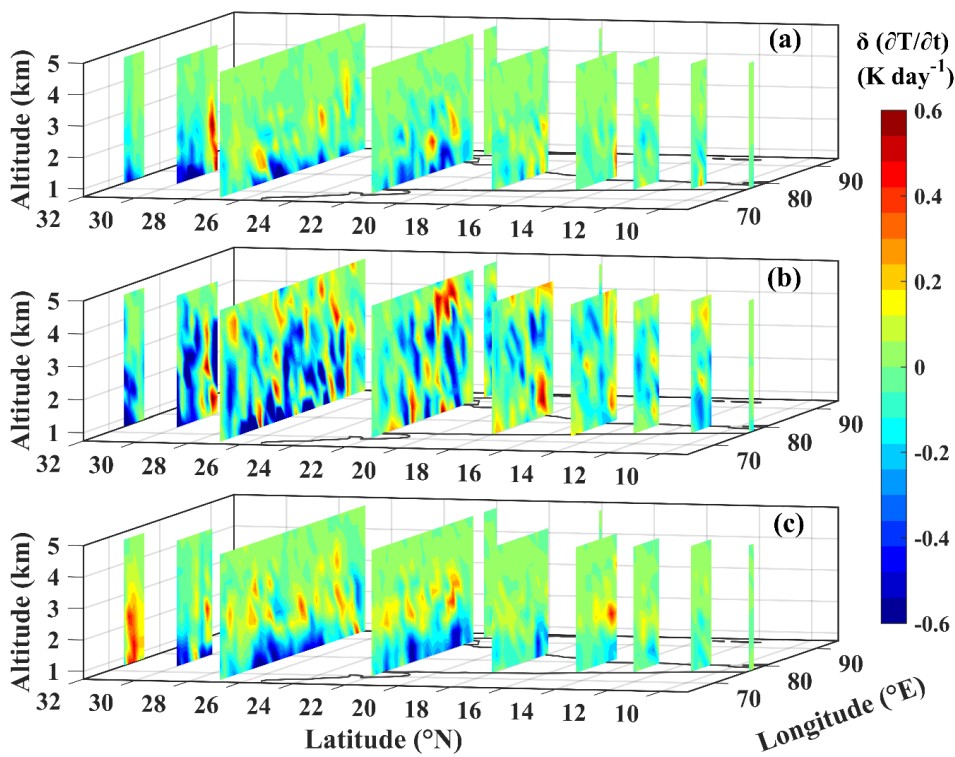


**Figure 12.** Difference in aerosol-induced atmospheric heating rate: $\delta(\partial T/\partial t) = [(\partial T/\partial t)_{asm} - (\partial T/\partial t)_{bg}]$ for (a) DJF, (b) MAM, and (c) ON seasons. The results reveal the realistic scenario over the Indian region of increased atmospheric warming due to BC aerosols at higher altitudes compared to the calculations carried out using satellite observations alone.





**Table 1.** Essential details of the aircraft and balloon measurements used in the present study. The station codes are written in parenthesis alongside each base station.

| Campaign | Mode | Year | Month | Base station | Reference |
|---|---|---|---|---|---|
| ICARB | Aircraft | 2006 | March | Bhubaneswar (BBR) | Babu et al. (2008); Moorthy et al. (2006; 2009) |
| | | | April | Chennai (CHN) Thiruvananthapuram (TVM) | |
| | | | May | Goa (GOA) | |
| WICARB | Aircraft | 2009 | January | Hyderabad (HDN) Vishakhapatnam (VSK) Mangalore (MGL) | Moorthy et al., 2010 |
| RAWEX | High altitude balloon | 2010 | March | Hyderabad (HDN) | Babu et al., 2011 |
| | | 2011 | January April | | |
| CAIPEEX-II | Aircraft | 2010 | October | Hyderabad (HDN) | Safai et al., 2012; Kulkarni et al., 2012 |
| | | 2011 | October November | | |
| RAWEX | Aircraft | 2012 | December | Ranchi (RNC) Jodhpur (JDP) Jaipur (JPR) Lucknow (LUK) Dehradun (DDN) Nagpur (NGP) | Babu et al., 2011; 2016; Moorthy et al., 2016 |
| | | | November December | Hyderabad (HDN) | |
| | | 2013 | April | Hyderabad (HDN) | |
| | | | April May | Nagpur (NGP) | |
| | | | May | Patna (PTN) Lucknow (LUK) Jaipur (JPR) Jodhpur (JDP) Dehradun (DDN) | |