# Peer review of "3D assimilation and radiative impact assessment of aerosol black carbon over the Indian region using aircraft, balloon, ground-based, and multi-satellite observations"

_EGUsphere, 2023_

## Author Comment (AC1)

**Author response to Reviewer #1 comments**

We sincerely thank the reviewer for the valuable comments. Based on the comments we received, careful modifications have been made to the manuscript. Our point-by-point response to the review comments are given below. The comments are marked in bold blue font and our responses are marked in normal black font below each comment.

**Reviewer #1**

**The manuscript titled '3D assimilation and radiative impact assessment of aerosol black carbon over the Indian region using aircraft, balloon, ground-based, and multi-satellite observations' by Kala et al. presents the three-dimensional gridded data set of black carbon (BC) aerosols over the Indian mainland utilizing data from the ground-based network, aircraft- and balloon-based measurements from multiple campaigns, and multi-satellite observations, followed by statistical assimilation techniques. It is a well-written work with excellent scientific merit, and I do not hesitate to recommend this manuscript for publication. I have a few minor suggestions below; those can be incorporated during the revision.**

We appreciate with thanks, the summary evaluation, and the positive recommendation.

**Specific comments:**

**Line 27-111: The introduction looks a bit wordy. Therefore, I recommend that the authors be concise in the introduction without losing the theme of this work.**

Complied with.

**Line 104-111: I would suggest removing these explanations here in the introduction since each section are well defined subsequently.**

Complied with.

**Line 16-18: Consider rewriting this sentence.**

Complied with. The sentence has been modified as: "The regional maps of BC-induced atmospheric heating derived using this dataset capture the elevated aerosol heating layers over the Indian region along with the spatial high over the Indo-Gangetic Plains." in Page no.1 [line no.16-18]).

**Line 20-23: Rephrase this sentence.**

Complied with. The sentence has been modified as: "This could strongly influence the upper tropospheric and lower stratospheric processes, including the vertical transport of BC to higher altitudes and thus have larger implications for atmospheric stability than what would be predicted using satellite observations alone." (Page no.1 [line no.20-23]).

**Line 32: I would suggest the authors add a few more references for the transport of BC aerosols to the high-altitude Himalayas in this context. E.g., Kompalli et al., 2016; Negi et al., 2019; Arun et al., 2019; Roseline et al., 2021; Gogoi et al., 2021; Arun et al., 2021;, etc. are the recent ones and authors can find more in the literature.**

Complied with, by adding a few more references. (Page no. 2 [line no.35-36])

**Line 41: faraway?**

Yes.

**Line 116-118: How are these seasons classified? Explain the criteria behind this.**

Based on the south-west monsoon which defines the climate of the region, there are generally four seasons considered over the Indian region. They are the cold and dry winter (December to February; DJF), pre-monsoon summer (March to May; MAM), monsoon (June

to September; JJAS) and post-monsoon (October to November; ON). In the present study, monsoon season has not been considered owing to the non-availability of sufficient aerosol measurements and the assimilated BC AAOD data, which form the base for processing the background BC absorption coefficient ($k_{bg}$).

**Line 124-127: Are the Aethalometer datasets pressure corrected for the high-altitude ground-based network stations? Explain these details also in the revised version.**

Yes, the high-altitude Aethalometer measurements have been pressure corrected. A sentence has been added in the revised manuscript as: "The Aethalometer operation and corrections carried out during different campaigns are explained in detail in the references listed in Table 1". (Page no.4 [ line no.124-125])

**Line 143: Explain the uncertainties about the Absorption AOD from OMI used in this study. Also, there are a number of caveats that concern the validity of the results in this study. I would recommend that the authors also add details about this in the discussions.**

Thank you for pointing it out. The assimilated absorption AOD, OMI absorption AOD, and their uncertainties are discussed in detail in Pathak et al. (2019). The main sources of uncertainty in the OMI AAOD are the sub-pixel cloud contamination caused by the considerably larger pixel size (1325 $km^2$), and the OMI retrievals overestimating AOD and underestimating SSA (Torres et al., 1998; 2005; 2007). Moreover, the assumptions on the vertical distribution of different aerosol species and the height of the aerosol layers contribute to the uncertainty in OMI retrievals. Torres et al. (2002) have estimated the total error in OMI AOD to be around 30% and that of SSA to be 0.05–0.1. Taking this into account, the present study has used the assimilated BC AAOD dataset generated by Pathak et al. (2019), having an uncertainty between 11% to 20% with a mean value of 15%, which is better than that of OMI AAOD.

- Pathak, H. S., Satheesh, S. K., Nanjundiah, R. S., Moorthy, K. K., Lakshmivarahan, S., and Babu, S. S.: 740 Assessment of regional aerosol radiative effects under the SWAAMI campaign–Part 1: Quality-enhanced estimation of columnar aerosol extinction and

absorption over the Indian subcontinent, Atmospheric Chemistry and Physics, 19, 11865-11886, https://doi.org/10.5194/acp-19-11865-2019, 2019.

- Torres, O., Bhartia, P. K., Herman, J. R., Ahmad, Z., and Gleason, J.: Derivation of aerosol properties from satellite measurements of backscattered ultraviolet radiation: Theoretical basis, Journal of Geophysical Research: Atmospheres, 103, 17099–17110, https://doi.org/10.1029/98JD00900.

- Torres, O., Decae, R., Veefkind, P., and de Leeuw, G.: OMI Aerosol Retrieval Algorithm, OMI Algorithm Theoretical Basis Document Volume III, Clouds, Aerosols, and Surface UV Irradiance, 4, pp. 47–71, 2002.

- Torres, O., Bhartia, P. K., Sinyuk, A., Welton, E. J., and Holben, B.: Total Ozone Mapping Spectrometer measurements of aerosol absorption from space: Comparison to SAFARI 2000 ground-based observations, Journal of Geophysical Research: Atmospheres, 110 (D10), https://doi.org/10.1029/2004JD004611, 2005.

- Torres, O., Tanskanen, A., Veihelmann, B., Ahn, C., Braak, R., Bhartia, P. K., Veefkind, P., and Levelt, P.: Aerosols and surface UV products from Ozone Monitoring Instrument observations: An overview, Journal of Geophysical Research: Atmospheres, 112 (D24), https://doi.org/10.1029/2007JD008809, 2007.

**Line155: How the BC AOD is estimated using OPAC model? Please elaborate more on this. Explain the constrained variables and uncertainties in doing so.**

The ground-based BC mass concentration measurements along with planetary boundary layer height data from Modern-Era Retrospective Analysis for Research and Applications-2 (MERRA2) are used to constrain the Mie scattering model Optical Properties of Aerosols and Clouds (OPAC), to obtain a near-realistic vertical variation of aerosols and thus the BC AOD. This is a good assumption for the vertical distribution of aerosols over the Indian region and is supported by several studies in the past across different seasons (Satheesh et al., 2008; Babu et al., 2010, 2011). These are already mentioned in Page 5 (lines 154 – 161). The uncertainty associated with the BC AAOD is within 11% to 20% with a mean value of 15% (Pathak et al., 2019).

- Babu, S. S., Moorthy, K. K., and Satheesh, S. K.: Vertical and Horizontal Gradients in Aerosol Black Carbon and Its Mass Fraction to Composite Aerosols over the East Coast of

Peninsular India from Aircraft Measurements, Advances in Meteorology, 2010, https://doi.org/10.1155/2010/812075, 2010.

- Babu, S. S., Moorthy, K. K., Manchanda, R. K., Sinha, P. R., Satheesh, S. K., Vajja, D. P., Srinivasan, S., and Kumar, V.: Free tropospheric black carbon aerosol measurements using high altitude balloon: do BC layers build "their own homes" up in the atmosphere?, Geophysical research letters, 38, https://doi.org/10.1029/2011GL046654, 2011.
- Pathak, H. S., Satheesh, S. K., Nanjundiah, R. S., Moorthy, K. K., Lakshmivarahan, S., and Babu, S. S.: 740 Assessment of regional aerosol radiative effects under the SWAAMI campaign–Part 1: Quality-enhanced estimation of columnar aerosol extinction and absorption over the Indian subcontinent, Atmospheric Chemistry and Physics, 19, 11865-11886, https://doi.org/10.5194/acp-19-11865-2019, 2019.
- Satheesh, S. K., Moorthy, K. K., Babu, S. S., Vinoj, V., and Dutt, C. B. S.: Climate implications of large warming by elevated aerosol over India, Geophysical Research Letters, 35, https://doi.org/10.1029/2008GL034944, 2008.

**Line 174: Consider rewriting this sentence.**

Complied with. The sentence has been modified as: "As part of various field campaigns, in-situ measurements of the vertical profiles of BC aerosols have been carried out at different locations and seasons over the Indian mainland and the adjoining oceans." (Page no.6 [line no.173-174]).

**Line 176, Figure-1: I recommend the authors to also show variabilities in the elevations of the study domain in the same map. There are different resources that provide digital elevation data.**

Thank you for the suggestion. Complied with and the revised Figure 1 is shown below. The elevation data is obtained from the United States Geological Survey Earth Resources Observation and Science (USGS EROS) archive.(Page no.4 [ line no. 119-121])

[Figure]

**Figure 1. Map showing the locations of surface (red markers) and aerial (aircraft and balloon) measurements. The colour scheme in the background shows the elevation data. The aircraft and balloon symbols respectively mark the aircraft and high-altitude balloon measurement locations. The station codes for the ARFI measurement sites are given in Table 1 and S1. (Page no. 25 [Line no. 897-900]).**

**Line 262: I suggest the authors to add some information regarding the role of crop residue burning during October-November in northern India.**

Complied with. A sentence on the role of crop residue burning during October-November in northern India has been added as: "During MAM, the IGP region is influenced by BC produced from agricultural residue burning and forest fires and the continental inflow of polluted aerosols to this region (Singh et al., 2019) whereas the crop residue burning has also been cited as a contributing cause during ON (Badarinath et al., 2006; Jain et al., 2014; Kaskaoutis et al., 2014)." (Page no. 8 [Line no. 262-265])

- Badarinath, K. V. S., Kiran Chand, T. R., and Krishna Prasad, V.: Agriculture crop residue burning in the Indo-Gangetic Plains—A study using IRS-P6 AWiFS satellite data. Current Science, 1085–1089, 2006.

- Jain, N., Bhatia, A., and Pathak, H.: Emission of air pollutants from crop residue burning in India. Aerosol and Air Quality Research, 14(1), 422–430, 2014, https://doi.org/10.4209/aaqr.2013.01.0031.

- Kaskaoutis, D. G., Kumar, S., Sharma, D., Singh, R. P., Kharol, S. K., Sharma, M.: Effects of crop residue burning on aerosol properties, plume characteristics, and long-range transport over northern India. Journal of Geophysical Research: Atmospheres, 119, 5424–5444, 2014, https://doi.org/10.1002/2013JD021357.

**Line 389, Figure-8: I would recommend the authors to show the $R^2$ values in these plots.**

Complied with: The modified plot with the $R^2$ values is shown below and in Page no. 30, line no. 923-927 in the revised manuscript)

[Figure]

**Figure 8. Scatter plots between $k_{obs}$ and $k_{bg}$ (top panels) and between $k_{obs}$ and $k_{asm}$ (bottom panels) for DJF (left panels), MAM (middle panels), and ON (right panels) seasons. The**

red line denotes the linear fit, the dashed blue line denotes the 1:1 line, and the scatter points are shown in gray. The equation of fit, correlation coefficient (R), $R^2$, and the number of scatter points (n) are shown in each sub plots.

**Line 418: Explain the statistical significance of the derived Heating rate values and their uncertainties.**

The uncertainty of the heating rate has been evaluated by first estimating its error covariance matrix, as shown below:

$$M = (I/(k-1)) (H*H^T) \tag{3}$$

Here, 'I' is an identity matrix of size n×n, 'H' is the heating rate matrix transformed into a matrix of size n-1, and 'k' is the number of time steps involved for each season. The background error covariance matrix is evaluated separately for DJF, MAM, and ON seasons. The final matrix 'M' consists of the variance of the heating rate for each grid, and the off-diagonal elements represent the co-variances. The uncertainty is estimated using the variance values (diagonal elements of matrix 'M'; Niu et al., 2008; Zhang et al., 2008; Singh et al., 2017; Pathak et al., 2019). The root mean square value of the uncertainties for the derived heating rate for all the seasons is found to be ~17%. The uncertainty associated with the estimated heating rate is mentioned in Page 12, line 415 – 416.

- Niu, T., Gong, S., Zhu, G., Liu, H., Hu, X., Zhou, C., and Wang, Y.: Data assimilation of dust aerosol observations for the CUACE/dust forecasting system, Atmospheric Chemistry and Physics, 8, 3473-3482, https://doi.org/10.5194/acp-8-3473-2008, 2008.
- Pathak, H. S., Satheesh, S. K., Nanjundiah, R. S., Moorthy, K. K., Lakshmivarahan, S., and Babu, S. S.: Assessment of regional aerosol radiative effects under the SWAAMI campaign–Part 1: Quality-enhanced estimation of columnar aerosol extinction and absorption over the Indian subcontinent, Atmospheric Chemistry and Physics, 19, 11865-11886, https://doi.org/10.5194/acp-19-11865-2019, 2019.
- Singh, R., Singh, C., Ojha, S. P., Kumar, A. S., and Kumar, A. K.: Development of an improved aerosol product over the Indian subcontinent: Blending model, satellite, and

ground-based estimates, Journal of Geophysical Research: Atmospheres, 122, 367-390, https://doi.org/10.1002/2016JD025335, 2017.

- Zhang, J., Reid, J. S., Westphal, D. L., Baker, N. L., and Hyer, E. J.: A system for operational aerosol optical depth data assimilation over global oceans, Journal of Geophysical Research: Atmospheres, 113, https://doi.org/10.1029/2009JD013364, 2008.

**Line 432: I recommend the authors add some literature related to aerosol-cloud interaction and the role of BC in it.**

Thank you for the suggestion. Complied with and the sentence has been modified in the revised manuscript with additional references on aerosol-cloud interactions as: "It could also lead to increased cloud activities in the higher atmosphere as BC (aged) can act as cloud condensation nuclei and influence the aerosol-cloud interactions (for e.g., Ackerman et al., 2000; Rosenfeld, 2000; Chuang et al., 2002; Penner et al., 2004; Kaufman et al., 2005; Lin et al., 2018; Zanatta et al., 2023)". (Page no. 12; line no. 433 - 436)

- Ackerman, A. S., Toon, O., Stevens, D., Heymsfield, A., Ramanathan, V., & Welton, E. (2000), Reduction of tropical cloudiness by soot, Science, 288, 1042-1047, https://doi.org/10.1126/science.288.5468.1042.

- Rosenfeld, D. (2000). Suppression of rain and snow by urban and industrial air pollution, Science, 287(5459), 1793–1796, https://doi.org/10.1126/science.287.5459.1793.

- Chuang, C. C., Penner, J. E., Prospero, J. M., Grant, K. E., Rau, G. H., & Kawamoto, K. (2002). Cloud susceptibility and the first aerosol indirect forcing: Sensitivity to black carbon and aerosol concentrations. Journal of Geophysical Research: Atmospheres, 107(D21), AAC-10, https://doi.org/10.1029/2000JD000215.

- Penner, J. E., Dong, X. Q., & Chen, Y. (2004). Observational evidence of a change in radiative forcing due to the indirect aerosol effect, Nature, 427(6971), 231–234, https://doi.org/10.1038/nature02234.

- Kaufman, Y. J., Koren, I., Remer, L. A., Rosenfeld, D., & Rudich, Y. (2005). The effect of smoke, dust, and pollution aerosol on shallow cloud development over the Atlantic Ocean, Proc. Natl. Acad. Sci., 102(32), 11207–11212, https://doi.org/10.1073/pnas.0505191102.

- Lin, L., Xu, Y., Wang, Z., Diao, C., Dong, W., & Xie, S. P. (2018). Changes in extreme rainfall over India and China attributed to regional aerosol-cloud interaction during the late

20th century rapid industrialization. Geophysical Research Letters, 45(15), 7857-7865, https://doi.org/10.1029/2018GL078308.

- Zanatta, M., Mertes, S., Jourdan, O., Dupuy, R., Järvinen, E., Schnaiter, M., Eppers, O., Schneider, J., Jurányi, Z., & Herber, A. (2023). Airborne investigation of black carbon interaction with low-level, persistent, mixed-phase clouds in the Arctic summer. Atmospheric Chemistry and Physics, 23, 7955–7973, https://doi.org/10.5194/acp-23-7955-2023.

**Line 475: Why authors offer BC absorption coefficient profiles at a horizontal resolution of 1°×1° and a vertical resolution of 0.5 km. What is the reason behind choosing this particular resolution? Is it possible to have a much higher-resolution dataset in this study?**

The data resolution is set to match the background data horizontal resolution of 1°×1°. The background data are chosen to have a vertical resolution of 0.5 km, which matches the vertical resolution of the observational data. It will not be beneficial to attempt for a higher-resolution (than that of the individual datasets used for the assimilation) dataset.

---

## Author Comment (AC2)

**Author response to Reviewer #2 comments**

We sincerely thank the reviewer for the valuable comments. Based on the comments we received, careful modifications have been made to the manuscript. Our point-by-point response to the review comments are given below. The comments are marked in bold blue font and our responses are marked in normal black font below each comment.

**Reviewer #2**

**This paper attempts to generate 3D composite grid data of BC from data obtained from various observation platforms and estimate the radiative effects of BC considering the vertical profiles. This study potentially provides materials with some implications to improve our understanding of BC over India, which can lead to a solid contribution to ACP. I am afraid, however, that there is critical issues need to be addressed before this paper can be considered for publication.**

We appreciate the summary evaluation and the positive comments.

**Major comments:**

**#1**
**In the earth sciences, assimilation generally refers to the process of integrating different types of observational data into a numerical model in order to improve the accuracy and reliability of model predictions and simulations. Even though the authors are using the mathematical methods used for assimilation, in this study they just combined the various observations together to create composite data. Using the word assimilation in the title and texts is misleading to the reader (and I misunderstood it too). I suggest using a different word.**

We appreciate your valuable feedback. However, we would like to retain the title as such because the mathematical framework employed in the present study is similar to the statistical assimilation studies involving numerical models. As such, we believe that the term 'assimilation' could be used in this context as well, which is further in line with the data and

methodology followed in Pathak et al. (2020) as well. However, we emphasize that this is different from the dynamical assimilation methods used in weather predication and climate models.

- Pathak, H. S., Satheesh, S. K., Moorthy, K. K., & Nanjundiah, R. S. (2020). Assessment of regional aerosol radiative effects under the SWAAMI campaign–Part 2: Clear-sky direct shortwave radiative forcing using multi-year assimilated data over the Indian subcontinent. Atmospheric Chemistry and Physics, 20(22), 14237-14252, https://doi.org/10.5194/acp-20-14237-2020.

**#2**

**It is difficult to understand the process of creating composite data from the description in the text and Figure 3. First, each of the data used should be described, and then the process shown in Fig.3 should be described carefully and in sequence.**

Thank you for the comment. The flow chart has been modified and the revised figure 3 is shown below and in Page no. 26 (line no. 904-906) in the revised manuscript. Each data set and the methodology are explained in Sect. 2, along with the references discussing them in detail.

[Figure]

**Figure 3. Flowchart describing the various data sets and steps involved in the data assimilation. $k_{obs}$, $k_{bg}$, and $k_{asm}$ respectively represent the observational, background and assimilated aerosol absorption coefficients.**

**#3**

**The authors are using the assimilation method in the wrong way. Figure 3 shows that common data (ARFINET BC AAOD, CALIPSO profiles) is used to generate both k_obs and k_bg. In other words, k_obs and k_bg are not independent. This does not satisfy the preconditions for maximum likelihood estimation, which is the basis of the variational method. In this case, the covariance between k_obs and k_bg must be taken into account.**

Thank you for the comment, which along with the previous comment have been considered to revise the flow chart (Figure 3) to be more comprehendible. It can be seen from the modified flowchart (shown in the previous comment) that $k_{obs}$ is generated using aircraft and balloon profiles measured as part of various campaigns and AFRINET BC AAOD (weighted with CALIOP profiles). On the other hand, $k_{bg}$ is generated merely using the assimilated BC AAOD developed in Pathak et al. (2019) weighted with CALIOP profiles and do not include in-situ measured profiles. Pathak et al. (2019) have used near-surface BC measurements, MERRA-2 planetary boundary layer height, dust AAOD estimated using Infrared Difference Dust Index, and OMI AAOD to generate this assimilated BC AAOD product but the airborne measurement data have not been used . These two different data sets have been used in the construction of background and observational data in the present study. Hence, we believe that the background and observational data can be considered to be independent.

**Specific comments:**

**#1**

**Line 141: It is very difficult to determine the absorption of dust alone from satellite observations. Please discuss the uncertainty of that and the uncertainty in the BC AAOD obtained.**

The uncertainty associated with BC AAOD has been estimated to be varying from 11% to 20% with a mean value of 15% (Pathak et al., 2019). Estimation of dust AAOD using multiple data sets is explained in the second paragraph of Sect. 2.2. and extended in the penultimate paragraph of Sect. 2.4 in the revised manuscript.

**2**

**Line 223: I am concerned about the very simple method of calculating background error covariance. Because the matrix A is only a deviation from the climatic value not including information of uncertainty of k_bs. Did you not try the method of calculating from the uncertainty of the data sets used to generate K_bg? I also concerned that the simple multiplication of the deviations (i.e., equation (3)) can properly estimated the off-diagonal components (i.e., the covariance in the spatial direction) of matrix B. Have you examined the structure of the covariance closely? This also relates to the advantage of the 3D-Var as pointed out in line 367-369.**

Thank you for these comments. The background error covariance can be derived either from climatological data or forecasts (Lewis et al., 2006). We have estimated the covariance matrix from climatological data (Eq. 3) as it effectively captures the spatial covariance structure (Pathak et al., 2019). Due to the relatively short data availability duration, it can be safely assumed that there are no discernible increasing or decreasing trends in the absorbing aerosol loading during the assimilation period. Thus, the deviations in monthly mean BC absorption coefficient across different years is considered as anomalies which are further employed for co-variance estimation. The square root of diagonal elements of co-variance matrix provides the estimates for the uncertainties in the background data at respective locations, while off-diagonal elements signify the cross-covariance values, which provide valuable insights on how the aerosol emissions from a grid influence the neighboring grids, and vice versa. It should be noted here that the present methodology does not employ any external estimates of the uncertainties in the background data.

Following up with reviewer R#2 suggestion, we have carefully examined the spatial covariance (off-diagonal components) within the error covariance matrix. For explanation, we have selected two representative locations, one each from North India and Peninsular India, at an altitude of 2 km amsl, for MAM season. The results, representatives of which are shown in Figure S4 in the supplementary section (Page no. 2 – 3; line no. 12 - 41), indicate that the covariance matrix adequately captures the spatial covariance for nearby locations. In the top panel of Figure S4, around the source region from peninsular India (marked by black diamond symbols), the covariance is high at an altitude of 2 km, indicating that the nearby grids are getting strongly influenced by the source region. Similarly, in the bottom panel of Figure S4,

where a source from north India is considered, high covariance is observed over the Indo-Gangetic Plain at an altitude of 2 km. A similar pattern of high covariance is observed for other source locations as well (plots are not shown).

[Figure]

**Figure S4. Spatial variation of the covariance between a single grid (marked with black diamond and arrow) and rest of the grids for peninsular (top panel) and northern (bottom panel) India.**

- Lewis, J. M., Lakshmivarahan, S., and Dhall, S.: Dynamic data assimilation: a least squares approach, Vol. 104, Encyclopedia of Mathematics and its Applications, Cambridge University Press, Cambridge, 2006.
- Pathak, H. S., Satheesh, S. K., Nanjundiah, R. S., Moorthy, K. K., Lakshmivarahan, S., and Babu, S. S.: Assessment of regional aerosol radiative effects under the SWAAMI campaign–Part 1: Quality-enhanced estimation of columnar aerosol extinction and absorption over the Indian subcontinent, Atmospheric Chemistry and Physics, 19, 11865-11886, https://doi.org/10.5194/acp-19-11865-2019, 2019.

**#3**

**Line 360: The values of Delta_k in Figure 7 show very fine-scale variation, particularly**

**in MAM. What is the cause of this? As a result, we also see fine-scale spatial variation in k_asm compared with Assimilated BC AAOD (Figure 4). This fine-scale variation is a realistic result?**

The assimilation of $k_{obs}$ with $k_{bg}$ has brought the fine-scale variation in the vertical profiles of $k_{asm}$ (because of incorporating realistic/measured profiles in the assimilation process) and this is reflected in $\delta k$. Such fine scale variations have been reported over the Indian region in several earlier studies (cited in the manuscript) and their signature in the assimilated BC AAOD data set is a realistic result. In fact, it is an important outcome of the study and fills a long-standing research gap, which would be helpful in improving the climate model simulations over the Indian region.

**#4**

**Line 367-369: In the method of obtaining from deviations as in equation (3), apparent correlations may appear, especially between grids separated by a distance. Have you examined about this?**

Thank you for pointing it out. We have examined this and observed that a noticeable apparent correlation exists between the grids over western India and the Indo-Gangetic plain during the MAM season. This correlation could be attributed to a composite effect resulting from long-range dust transport as well as strong BC emissions occurring over the entire Indian region. However, after considering the absence of such a prevalent correlation in other regions and seasons, we have determined that its impact on the assimilation process is not expected to be detrimental.

**#5**

**Line 387-394: It is obvious that K_asm is more consistent with K_obs than K_bg; the discussion using Fig.8 makes no sense. If authors want to verify Kasm, authors should do so with independent data.**

We acknowledge the significance of validating $k_{asm}$ (assimilated data) using independent sources. However, it is important to note that in our study, the availability of $k_{obs}$ (observed

data) for validation purposes is limited. The scarcity and sparsity of $k_{obs}$ pose serious challenges in reserving a specific subset solely for validation, as it would largely reduce the already limited data available for assimilation. Considering these constraints, our intention in Figure 8 was to investigate the extent to which the assimilation process improved the correlation between $k_{bg}$ (background data) and $k_{obs}$. Nevertheless, we appreciate the reviewer suggestion to focus on validating $k_{asm}$ using independent $k_{obs}$. As such, we have performed the validation of $k_{asm}$ against independent $k_{obs}$. Among the 35 stations, we utilized $k_{obs}$ data from 25 stations for assimilation and reserved $k_{obs}$ data from 10 stations for validation. To ensure statistical significance and an adequate number of data points in the analysis, the validation datasets from all three seasons were consolidated. The results from this analysis are shown in Figure RC2. It can be observed that after assimilation, using the new data subsets, $k_{obs}$ still aligns more closely with $k_{asm}$ as compared to $k_{bg}$, highlighting the robustness of our methodology.

[Figure]

**Figure RC2. Scatter plots between profiles of (a) $k_{obs}$ and $k_{bg}$ and (b) $k_{obs}$ and $k_{asm}$. The red line denotes the linear fit, the dashed blue line denotes the 1:1 line, and the scatter points are shown in gray. The equation of fit, correlation coefficient (R), and the number of scatter points (n) are shown in each sub plots.**

**Line 413-425: Very interesting point. If there are any previous studies that make similar points, please include them.**

Thank you. The vertical profiles of aerosol single scattering albedo and the possibility of obtaining more realistic aerosol radiative forcing profiles were shown in Vaishya et al. (2018) and Manoj et al. (2020) using aircraft measurements (limited temporally and spatially to smaller domains as compared to the present study) and are already cited in the manuscript.

- Manoj, M.R., Satheesh, S.K., Moorthy, K.K. and Coe, H.: Vertical profiles of submicron aerosol single scattering albedo over the Indian region immediately before monsoon onset and during its development: research from the SWAAMI field campaign. Atmospheric Chemistry and Physics, 20(6), pp.4031-4046, https://doi.org/10.5194/acp-20-4031-2020, 2020.

- Vaishya, A., Babu, S. S., Jayachandran, V., Gogoi, M. M., Lakshmi, N. B., Moorthy, K. K., and Satheesh, S. K.: Large contrast in the vertical distribution of aerosol optical properties and radiative effects across the IndoGangetic Plain during the SWAAMI–RAWEX campaign, Atmospheric Chemistry and Physics, 18, 17669-17685, https://doi.org/10.5194/acp-18-17669-2018, 2018.

**#7**

**LIne437-450: Were you able to find any traces of self-lofting in this data set?**

No, the self-lofting of aerosols occurs due to processes associated with relatively lower time scales as compared to the time intervals used in this study to generate the assimilated data set.

**#8**

**Line426-470: Although related to the need for an accurate vertical profile of BC, it is mostly redundant as it is mostly a description of previous studies. The description should be more concise in conjunction with the results of this study.**

Complied with.